# Parental Resources in Parents of Children with Special Needs (SNs) at the Time of COVID-19

**DOI:** 10.3390/jcm12020475

**Published:** 2023-01-06

**Authors:** Ambra Gentile, Concetta Polizzi, Giulia Giordano, Sofia Burgio, Marianna Alesi

**Affiliations:** Department of Psychology, Educational Sciences and Human Movement, University of Palermo, 90128 Palermo, Italy

**Keywords:** parental burnout, parenting sense of competence, COVID-19 restrictions, learning disability

## Abstract

Background. The limitations imposed by governments for containing the spread of COVID-19 have affected familial relationships, especially those of families dealing with children with special needs or chronic illness conditions. The current study aims to better understand what pathological/disability condition has impacted parental resources, sense of competence, and perception of children’s executive functioning the most. Methods. A sample of 648 parents was asked to answer a survey assessing children’s condition (typical development, specific learning disorder, autism spectrum syndrome, chronic illness), parental resources, parenting sense of competence (distinguished into parental satisfaction and self-efficacy), and parents’ perception of their children’s executive functioning. A MANOVA model was performed to assess differences in parental resources, sense of competence, and perception of the children’s executive functions according to their condition. A path analysis model was performed to examine the impact of sense of competence and children’s condition on parental resources and children’s executive functions. Results. Parents with children with specific learning disorder reported lower scores of parental resources in terms of total scores, common antecedents, and specific antecedents compared to parents with typically developed children (total scores: MD = 56.45, *p* < 0.001; common antecedents: MD = 22.28, *p* < 0.00; specific antecedents: MD = 34.17, *p* < 0.001), parents with autistic children (total scores: MD = 62.79, *p* = 0.01; common antecedents: MD = 24.03, *p* = 0.01; specific antecedents: MD = 38.76, *p* = 0.02) and parents of children with chronic illness (total scores: MD = 37.42, *p* = 0.04; common antecedents: MD = 16.59, *p* = 0.01). The path analysis model shows a direct effect of parental satisfaction (β = 0.26) and pathology/disability (β = −0.10) on parental resources that, in turn, influence parents’ perception of their children’s executive functioning (β = 0.24). Conclusion. Although no data about the prepandemic situation is available, the current study highlights that parental stress influence children’s cognition. Moreover, parents of children with special needs seemed to be challenged during COVID-19, especially parents of children with specific learning disorders, that are already stressed out by their children’s condition. Therefore, academic services should undertake preventive measures to preserve parental well-being and to provide a supportive environment for children, especially for those with atypical development.

## 1. Introduction

The spread of the COVID-19 infection and the preventive measures of lockdown, such as social restrictions and closure of nonessential businesses, have produced many significant changes in families’ habits, with an increased risk for stress and detrimental effects on well-being of parents. To this day, few studies considered the impact of the pandemic on parents’ mental health, and this was particularly true for families with children with special needs (SNs)—i.e., intellectual disabilities (IDs), specific learning disabilities (SpLD), autism spectrum disorder (ASD), attention-deficit/hyperactivity disorder (ADHD), cerebral palsy (CP), genetic disorders—who experienced a detrimental effect in a disproportionate way compared to those with typically developing (TD) children [1,2,3].

The uncertainty and fears surrounding COVID-19 lead to a unprecedented variations in everyday life: strict rules on socialization and interaction, reduced availability of therapeutic and educational professional support (psychologists, therapists, counselors), uncertainty about financial standing, new working adjustments such as remote working, and the necessity to switch to home-schooling, all while managing household commitment and domestic responsibilities. All these elements have been found to contribute to high levels of parental stress with significant negative consequences for parents’ mental wellbeing [4,5,6,7,8].

The extant scientific literature examining the impact of these changes during the pandemic and their links to parental stress indicated a number of risk factors shown to increase the magnitude of stress levels perceived, the most salient found among mothers, people aged between 18 and 30 years, single parents, part-time workers or unemployed individuals, individuals having a previous history of trauma or disease, individuals with a profile oriented to neuroticism, individuals having a child with SNs, and individuals with infected relative [9,10,11]. Above all, parents of children with disabilities—who would normally experience significant challenges due to their child’s disability—have been shown to suffer the most from the social restrictions [12,13,14].

It is widely recognized that parents with children with SNs show higher rates of distress than those with typically developing children, which derives from their increased duty of care and combat of social stereotypes, which leads to experiencing isolation and feeling the need for added support [15,16,17,18]. In such circumstances, extreme levels of stress are likely to generate burnout in which an imbalance exists between parenting demands and available personal resources. Parental burnout develops when parental resources are not sufficient to cope with the parenting demands and the levels of parental stress increase in a global condition in which high rates of perfectionism, low emotional intelligence, maladaptive child-rearing practices, lack of support from the coparent or the enlarged family are present [19,20]. This can lead parents to manage their children’s education in a less sensitive way, to use less effective coping strategies, or to decrease their ability to face challenging tasks, with the increasing risk of exacerbating the children’s disability. During the pandemic, all these factors negatively impacted the quality of parenting and family relationships [21,22,23,24].

On the other hand, the sense of competence engendered by parents’ levels of satisfaction and self-efficacy should be another aspect of consideration when analysing parental resources and burnout. Several studies highlighted that parental burnout can be predicted by low levels of parenting self-efficacy [25,26,27,28]. A study by Chartier, Delhalle, Baiverlin, and Blavier [25] assessed the impact of COVID-19-related measures on parents’ satisfaction, reporting that parents who experienced stress at that time displayed lower levels of satisfaction than parents who reported experiencing no stress. Similarly, Moscardino, et al. [29] reported lower parental stress in cases displaying high parental self-efficacy. However, parents with disabled children were shown to experience increasing levels of burnout as their self-efficacy decreased [30].

On the other hand, concerns of parents with children of SNs were exacerbated by their negative perception of children’s cognitive abilities and, in particular, by the impaired perception of executive functions (EF) [19].

Executive functioning is an umbrella term involving a set of goal-directed cognitive functions [31], namely defined as working memory (the capacity to monitor, hold in mind, and perform items and mental representations); inhibition (the ability to control and repress an instinctive response if not suitable); shifting (the ability to move from one task or mindset to another one); fluency (the ability to name as many items as possible (e.g., words, colors) in a time range); and planning (the ability to apply and monitor a sequence of thoughts or actions in order to achieve a goal) [32].

In this context, the lack of an appropriate educational environment or an extended break from it is argued to enhance children’s levels of distress, and negatively impact inhibitory control, cognitive flexibility, planning, attention, and decision making. Conversely, the same negative effects could apply vis à vis the family unit.

Notably, children with SNs tend to report lower executive functioning compared to TD children. In addition, several studies regarding children with specific learning disorders, report lower executive functioning in terms of visuospatial working memory [33] and central executive functioning [34], whereas children with autism report difficulties in inhibition, planning, and cognitive flexibility [35].

During the lockdown phase, children with SNs displayed less ability to manage and execute tasks demanding executive functions [36]. It is possible that parents with experience with disability or medical problems felt less connected to their children and experienced increased daily distress and worries about life goals that may have, in exchange, negatively influenced their parenting actions. Therefore, this could lead to an increased chance for chaotic family relations, a diminished attitude toward planning or implementing cognitively challenging tasks, an increase in controlling behaviour over the children, an attitude to limit their autonomy in their own decision-making processes and the use of strategies to cope with stressful events. Moreover, mothers reporting higher levels of stress were more likely to show less sensitivity; a linking of their own negative emotions to their children’s behaviour and the establishment of more maladaptive interactions, with the consequence being triggering behavioral problems in their children [19,20]. The long-term result would be a supplementary impairment or delay in children’ executive functional development. This process could be interpreted in light of an ecological perspective demonstrating how the development of executive functions in their long-term are influenced by both macrocontextual (e.g., socioeconomic status and cultural contexts) and microcontextual (e.g., parenting skills, family members psychological variables, language, parent–child relationships) factors [37,38,39].

Given the absence of other studies conducted prepandemic and during the pandemic on the relationships between parental resources in different conditions (typical development, specific learning disorder, autism, and chronic illness), the objective of the current study is to fill this gap.

Therefore, the current study aims to assess differences between parents with typical development children and parents with atypical pattern of development children in terms of parental resources, parental satisfaction, self-efficacy, and executive functioning of their children. Specifically, the hypotheses of the study are as follows.

**Hypothesis 1 (H1).** 
*Parents of children with SNs have lower parental resources than parents of TD children.*


**Hypothesis 2 (H2).** 
*Parents of children with SNs have lower satisfaction and lower self-efficacy than parents of TD children.*


**Hypothesis 3 (H3).** 
*Parents of children with SNs perceive lower executive functioning quality than parents of TD children.*


A path analysis model will be performed to analyze the impact of parents’ satisfaction, self-efficacy, and children’s condition (typical development, specific learning disorder, autism, and chronic illness) on parental resources and their effects on perception of children’s executive functioning (working memory, attention, planning, shifting, inhibitory control).

## 2. Materials and Methods

### 2.1. Participants

The study consists of an online survey. The potential participants were recruited through social media and online advertisement immediately after the first wave of COVID-19 (from May 2020). Participants willing to take part in the survey were informed about the aims and procedures through a brief description of the study; the informed consent was expressed inside the survey, before filling out the questionnaire, which took 20–25 min to complete.

The study was conducted respecting the Declaration of Helsinki principles and was approved by the University of Palermo Bioethics Committee (no. 13/2020).

Participants were selected if they (a) were parents, (b) were at least 18 years old, and (c) had a child between 4 and 17 years of age.

The sample consisted of 648 participants, mainly Italians (99.2%), 586 of whom were mothers (90.4%) and 62 of whom were fathers (9.6%), with and age range between 36 and 45 years old (54.3%). The majority of the participants were married (84.6%), and few of them were separated or shared joint custody (6.8%) or cohabited (7.7%). Less than 1% consisted of single parents.

Concerning education, participants mainly held a degree (36.4%) or a high school diploma (32.7%), followed by professional diploma (12.2%), Ph.D. or specialization (9.6%), middle school diploma (8.0%), and primary school certificate (1.1%). Participants worked primarily as teachers (29.1%) and government employees (27.3%).

Parents referring to male children were 54.5% of the sample, and 45.5% referred to their children as female. The average age of the child was 10.5 years (±4.59), mostly first-born (70.2%) and second-born (24.1%). Concerning the developmental conditions of the children, 587 children presented a “typical” pattern of development and 61 were exhibiting an atypical pattern of development, including learning disabilities (23 children), autism (9 children), and chronic or severe pathologies (29 children).

### 2.2. Measures

#### 2.2.1. Sociodemographic Data

The survey consisted of several standardized measures. Sociodemographic data were collected through questions investigating gender, age, nationality, marital status, level of education, and work regimen during COVID-19 pandemic. The data concerning the children included gender and birth order, with the presence of pathologies/disabilities/disorder being assessed through the question “does your child have one or more of the following conditions?” and the answers were reported in the following way: TD (0), SpLD (1), ASD (2), and CI (3). Chronic illness is considered as a broader category collecting children with dysmetabolic syndromes, cardiopathologies, or other physical illness not related to intellectual or mental disabilities.

#### 2.2.2. Parental Burnout

Parental burnout was assessed by analysing the balance between risks and resources—BR^2^ [40], and a 39-point self-report item questionnaire was deployed to investigate parental balance between risks (parental distress-enhancing factors) and resources (parental distress alleviating factors). The questionnaire considers bipolar items on 11 levels, ranging from −5 to +5. Out of 39 items, 14 items defined common antecedents as risk factors, indicating predictors of and parental burnout and burnout related to working conditions (e.g., “It is difficult for me to reconcile my family life and my professional life”) whereas 25 items defined specific antecedents showing aspects strictly related to parental burnout (e.g., “Due to my parenting responsibilities, I can never find time for myself”). The total score ranged between −195 and +195. A negative score indicates that parents’ level of risk is higher than their resources, and a positive score indicates a higher prevalence of resources over risk factors. A score of zero indicates an equal level of risk and resources. The reliability of the scale was excellent (Cronbach’s alpha for global scale α = 0.96; common antecedents: α = 0.88; specific antecedents: α = 0.94).

#### 2.2.3. Parenting Sense of Competence

Parenting sense of competence was evaluated through the Parenting Sense of Competence Scale [41,42]. The scale consists of 16 self-reported items on a 6-point Likert scale assessing parental satisfaction (“Even though being a parent could be rewarding, I am frustrated now while my child is at his/her present age”) and parental self-efficacy (“Being a parent is manageable, and any problems are easily solved”). Higher scores indicate a higher sense of competence. The reliability of the scale was good (Cronbach’s alpha for global scale: α= 0.86; parental self-efficacy: α = 0.77; parental satisfaction: α =0.77).

#### 2.2.4. Parental Perception of Children’s EF

The parental perception of children’s EF was measured through the executive functioning self-report(EF-SR) [43]. The scale consisted of 20 items concerning perceptions about the child’s cognitive abilities in relation to the environmental conditions, such as working memory (e.g., “My child is not good at remembering sequences of items, for example, numbers or words”), attention (e.g., “My child has difficulty ignoring extraneous thoughts when he/she performs a task”), shifting (e.g., “My child has difficulty moving from one task to another (for example from a math task to a science task”), planning (e.g., “My child has troubles performing tasks that have more steps”) and inhibitory control (e.g., “My child has difficulty with concentration while working in the classroom”). The items were evaluated on 4-point Likert scale ranging from 1 (always) to 4 (never). Higher scores indicated minor difficulties on EF tasks. A total score (from 20 to 80) and subscores (from 4 to 16) for each area were obtained. The reliability for the global scale was excellent (Cronbach’s alpha α = 0.95) and good for the subscales (working memory: α = 0.87; attention: α = 0.82; planning: α = 0.76; shifting: α = 0.85; inhibitory control: α = 0.80).

### 2.3. Data Analysis

Data were analysed through the Statistical Package for the Social Sciences (SPSS; IBM, version 24). Descriptive statistics of the considered population were presented, distinguishing the scores between parents of children having a chronic pathology and parents having ASD children or with learning disability. A multivariate analysis of variance (MANOVA) model was conducted to detect differences among the healthy children group, the children with CI group and the autism/learning disability children group. Finally, a path analysis model has been performed with Mplus 7 [44] to analyze the Impact of parents’ satisfaction, self-efficacy, and children’s condition (TD, SpLD, ASD, and CI) on parental resources and their effects on perception of children’s executive functioning (working memory, attention, planning, shifting, inhibitory control).

## 3. Results

### 3.1. Sociodemographic Data

Percentages for marital status, level of education, and work regimen during the COVID-19 pandemic were reported distinguishing between TD children, children with SpLD, children with CI, and children with ASD (Table 1).

No differences were found in marital status and work regimen during the pandemic, but significant differences emerged concerning education level. All the MANOVA models were first performed to analyse parents’ education level and occupation, but as the results revealed that all the respondents were educated at the university level and were mainly employed in either the education or governmental sectors, these variables were removed because it was found that they were not influential on the outcomes analysed in this study.

### 3.2. Parental Risk and Resources

Table 2 contains the score distribution for parents of TD children, children with SpLD, children with CI, and children with ASD.

The MANOVA analysis highlighted a significant difference among the four groups (F_3644_ = 6.49, *p* < 0.001) on the total burnout score, as well as on the common antecedents (F_3644_ = 5.63, *p* = 0.001) and specific antecedents (F_3644_ = 6.50, *p* < 0.001).

The post-hoc test with LSD correction showed that parents with children with learning disabilities reported lower resources than parents with typically developed children, children with ASD, and children with CI.

Specifically, parents with children with learning disabilities reported lower burnout total scores than parents with typical development children (MD = 56.45, *p* < 0.001), with children with ASD (MD = 62.79, *p* = 0.01), and with CI (MD = 37.42, *p* = 0.04).

Similarly, parents with children with learning disabilities reported lower scores than parents with typical development children (MD = 22.28, *p* < 0.001), with ASD children (MD = 24.03, *p* = 0.01), and with CI (MD = 16.59, *p* = 0.01) on common antecedents, and reported lower scores than parents with typical development children (MD = 34.17, *p* < 0.001) and with ASD children (MD = 38.76, *p* = 0.02) concerning specific antecedents.

### 3.3. Parenting Sense of Competence

Table 3 shows the dimensions of parenting sense of competence across the three groups.

The MANOVA model showed no differences among the three groups on the two dimensions of parenting sense of competence (parental satisfaction: F_2646_ = 0.32, ns, parental self-efficacy: F_2646_ = 0.11, ns).

### 3.4. Parental Perception of Children’s EF

Table 4 reports the scores for parental perception of children’s EF.

The MANOVA model showed significant differences among the four groups concerning working memory (WM; F_3638_ = 2.81; *p* = 0.04), attention (F_3638_ = 3.96; *p* = 0.01), and shifting (F_3638_ = 2.72; *p* = 0.02).

From the post-hoc with LSD correction, working memory was significantly worse according to parents of children with learning disabilities than those with typical development (MD = 1.53; *p* = 0.03), and parents of ASD children compared to parents with typical development children (MD = 2.16; *p*= 0.05). Similarly, attention was significantly worse according to parents of learning disability children than those with typical development (MD = 1.40; *p* = 0.03), according to parents with ASD children compared to parents with typical development children (MD = 2.72, *p*= 0.007), and according to parents with ASD children compared to parents with children affected by CI (MD = 2.54, *p* = 0.02). Parents with children with autism also reported lower scores on shifting compared to parents of CI children.

### 3.5. Predictors of Parental Resources and Perception of Children’s EFs

No collinearity among parental satisfaction, parental self-efficacy, and parental resources in predicting parental perceptions of children’s executive functions was retrieved. We calculated the correlation among the overmentioned variables by using Pearson’s r (Table 5). Parental resources are positively related to parental satisfaction (r = 0.20), parental efficacy (r = 0.09), and parents’ perception of their children’s executive functions (r = 0.32). Perception of children’s EF is positively related to parental satisfaction (r = 0.44) and parental efficacy (r = 0.37).

A path analysis model (see Figure 1) was hypothesized to test parental self-efficacy, parental satisfaction, and pathology (0 = TD, 1 = SpLD, 2 = ASD, 3 = CI) concerning parental resources, related, in turn, to executive functioning. The model showed an excellent fit (chi-square fit = 3473.89, df = 42, *p* < 0.001; CFI: 0.98, TLI: 0.97; RMSEA: 0.06, 90% CI 0.04, 0.07). Parental efficacy (β = 0.26) and presence of pathology (β = −0.10) are significantly related to parental resources, whereas parental resources are significantly related to the perception of EFs of their children (β = 0.24). Parental self-efficacy did not influence parental resources but showed a good positive relation with parental satisfaction (β = 0.67) and parents’ perception of children’s EF (β = 0.37), which is positively related to parental satisfaction as well (β = 0.42). The presence of pathology showed no relationships with parental satisfaction, parental self-efficacy, and parents’ perception of children’s EF.

## 4. Discussion

The aim of the current study was to assess the differences between parents with children with typically developed children, children with special needs, and children with chronic illness, in terms of parental resources, parenting sense of competence (parental self-efficacy and satisfaction), and perception of their children’s executive functioning in the period in which COVID-19 restrictions were enforced. Specifically, we hypothesized that parents with children with SNs would have fewer resources, less satisfaction, and less efficacy than parents of children. Moreover, parents with children with SNs would perceive lower executive functioning quality than parents of children. Finally, an explorative model to assess the impact of sense of competence and the children’s condition (typical development, specific learning disorder, autism, chronic illness) on parental resources and parents’ perception of children’s executive functioning was performed.

Our hypotheses were partially confirmed. As previously mentioned, it is widely recognized that parents with children with SN show higher distress rates than families with typically developing children because of the high care responsibilities, challenges, social stereotypes, and loneliness [15,16,17,18]. In our study, parents of children with learning disorders reported lower levels in common antecedents, specific antecedents, and total scores compared to parents of typically developed children, children affected by autism and by chronic illness. This result is not found to be surprising: children with learning disorders attend the same school classes as typically developed children, with the difference that the teacher adapts each school program considering the impairments in a specific learning area [45], and the same teaching methodology was maintained in the distance learning for COVID-19. Indeed, according to the Italian Law on Learning Disabilities (DL 170/2010), the majority of students with learning disorders had a specific personalized didactic plan (PDP) at school, consisting of dispensatory measures (e.g., reduction and adaptation of homework) and compensatory instruments (e.g., using the computer for writing, use of a calculator or concept maps during tests). In this way, parents of children with learning disorders could have faced difficulty in managing their children’s homework. Indeed, as reported by Marchese, Grillo, Mantovani, Gargano, Limone, and Indrio [45], distance learning imposed by the pandemic severely impacted children suffering from learning disorders and their parents, who declared feeling inadequate in fulfilling their children’s special needs. Conversely, children with autism had the support of a special education teacher that was maintained with distance learning for the COVID-19 period.

Parenting sense of competence was assessed in terms of parental self-efficacy and parental satisfaction, but no significant differences emerged among conditions. This result is not in line with the extant literature, which reports instead that parents of children with disability and SNs usually have lower levels of self-efficacy and satisfaction [46,47,48]. A study by Whitley, et al. [49] found low levels of parental self-efficacy in parents of children with SNs. These parents were not satisfied with the academic support received from the school. We did not assess this variable in our sample; therefore, we cannot exclude that the parents in our sample received adequate support from the school, and, for this reason, their scores could be similar to the ones of parents with typically developing children. Moreover, parental satisfaction seems to be lower in the case of children’s severe disability. Again, the severity of children’s disability or disorder did not form part of our analysis, and hence it cannot be excluded that the sample analysed might have involved milder or less prominent disabilities. Finally, even though we expected lower levels of self-efficacy in children with atypical development, as reported by Seo and Kim [30], parental self-efficacy lowered also in families with typically developing children.

It is well-known that children with SNs report lower levels of executive functioning than children with typical development [35]. In this study, significant differences emerged in parental perception of children’s executive functioning where parents with children with specific learning disorders observed lower levels of working memory and attention compared to parents of children with typical development, whereas parents with children with autism observed lower levels in terms of attention and shifting, respectively, compared to parents with typically developed children and to parents with chronic illness children. These results are in line with our expectations and are backed up by results of previous studies examining the detrimental impact of COVID-19 on children’s executive functioning [50] also in the case of typical development and SNs [51], but as no prepandemic data were collected, we have no comparative measure to clearly indicate the detrimental impact this has on the factors analysed. Nonetheless, the intent of this research was to analyse how the severe restrictions imposed during first wave by the Italian government may have impacted parenting burnout, parental self-efficacy and satisfaction and the presence of pathology on the perceived EFs.

A path analysis model was then designed, in which a direct effect of parental satisfaction and children’s condition on parental resources emerged, that were positively related to children’s executive functions. Similarly, other studies have examined the relationship between parental stress and parental satisfaction during COVID-19, finding that lower levels of satisfaction reflect on children’s well-being [52]. Other studies have reported that the interaction between parents and children during the lockdown directly impacted children’s EFs [53,54].

However, in our sample, parental self-efficacy was not related to parental resources or their children’s condition. This result contradicts the other contributions investigating the relationship between self-efficacy, parental stress, and resources and children’s condition, especially in cases of severe disability [55] and specific learning disorders [56]. Finally, when screening the data for parents’ education level and parents’ occupation, we did not find any significant influence on the outcome variables of this study, probably because the lockdown measures could have changed the familiar assets independently from the familiar socioeconomic background.

The current study represents a description of parenting difficulties and challenges during COVID-19, especially for families with atypical development children. It has the advantage of detecting these aspects on a big sample, and it compares parents whose children are in different conditions. In this way, the study fills a gap in the literature concerning children’s disability and, in particular, special needs [1,57]. However, some limitations are identified: first, the sample mainly consisted of typically developing children, and this was because the survey was shared with the general population without targeting parents in a specific condition. Secondly, the study is correlational; therefore, we cannot be sure that some external variables (such as the presence of a support teacher and perception of academic support) could partially explain our results. Moreover, the survey was distributed online, thus excluding individuals with no Internet access. In addition, we detected another source of response bias: the respondents were mainly females, who are found to be more likely to answer surveys, especially online [58]. Finally, because no previous data about the considered variables were detected, we cannot be sure that the pandemic worsened parents’ conditions.

To sum up, given that parents of children with SNs, especially those having specific learning disorders children, faced difficulties in managing COVID-19 restrictions with higher levels of parental stress, we can hypothesize that improving school academic services in a pandemic scenario could be useful to preserve parental well-being and to provide a supportive environment for children. From parents’ perspective, maintaining or supplying professional help during the COVID-19 pandemic could play a key role in preventing regression and worsening of their child’s skills and behaviors. Future research should focus on better identifying the variables that determine parental burnout in critical situations, such as the pandemic, with special consideration for parents with children with SNs.

## Figures and Tables

**Figure 1 jcm-12-00475-f001:**
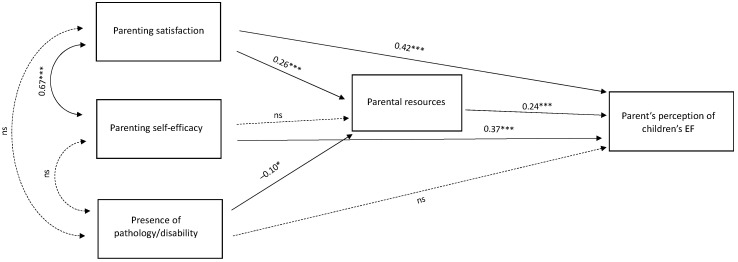
Path analysis model. *** *p* < 0.001, * *p* < 0.05.

**Table 1 jcm-12-00475-t001:** Sociodemographic data distinguished for families with typical development (TD), specific learning disorder (SpLD), with autism spectrum disorder (ASD), and chronic illness (CI).

	TD	SpLD	ASD	CI	χ^2^	*p*
Marital Status					14.67	0.10
Married	500 91.2%	18 3.3%	6 1.1%	24 4.4%		
Divorced/separated	38 86.4%	2 4.5%	3 6.8%	1 2.3%		
Cohabiting	44 88.0%	3 6.0%	0 0.0%	3 6.0%		
Single parents	5 83.3%	0 0.0%	0 0.0%	1 16.7%		
Education Level					27.00	0.03 *
Primary School	5 71.4%	0 0.0%	0 0.0%	2 28.6%		
Middle School	43 82.7%	3 5.8%	1 1.9%	5 9.6%		
Professional Diploma	72 91.1%	1 1.3%	2 2.5%	4 5.1%		
High-school Diploma	187 88.2%	12 5.7%	2 0.9%	11 5.2%		
Degree	222 94.1%	5 2.1%	2 0.8%	7 3.0%		
PhD	58 93.5%	2 3.2%	2 3.2%	0 0.0%		
Work regimen during pandemic					21.26	0.63
Smartwork	222 90.6%	12 4.9%	3 1.2%	8 3.3%		
Temporarily suspended	99 91.7%	3 2.7%	0 0.0%	6 5.6%		
Went to the workplace	150 92.0%	4 2.5%	2 1.2%	7 4.3%		
Lost the job	18 99.0%	1 0.5%	0 0.0%	1 0.5%		
Prefer not to answer	97 87.3%	3 2.7%	4 3.6%	7 6.4%		

* *p* < 0.05. Legend: TD, typical development; SpLD, specific learning disorder; ASD, autism spectrum disorder; CI, chronic illness.

**Table 2 jcm-12-00475-t002:** Scores of parental risk and resources.

Children BR^2^ Scores	TD (*n* = 587)		SpLD (*n* = 23)		ASD (*n* = 9)		CI (*n* = 29)		Statistics	
	Mean	SD	Mean	SD	Mean	SD	Mean	SD	*F* Value	η^2^_p_
Total score	70.88	62.99	14.43	68.91	77.22	62.20	51.86	64.55	6.49 ***	0.029
Common antecedents	21.59	23.80	−0.70	27.90	23.33	23.01	15.90	28.62	6.68 ***	0.030
Specific antecedents	49.30	41.89	15.13	44.07	53.89	43.97	35.97	48.20	5.63 **	0.026

*** *p* < 0.001, ** *p* < 0.01. Legend: TD, typical development; SpLD, specific learning disorder; ASD, autism spectrum disorder; CI, chronic illness.

**Table 3 jcm-12-00475-t003:** Scores of parenting sense of competence.

Children Parenting	TD (*n* = 587)		SpLD (*n* = 23)		ASD (*n* = 9)		CI (*n* = 29)		Statistics	
	Mean	SD	Mean	SD	Mean	SD	Mean	SD	*F* Value	η^2^_p_
Satisfaction	33.55	9.34	33.04	7.47	33.67	10.82	35.21	9.21	0.32	0.001
Efficacy	27.22	6.72	26.87	6.17	27.44	8.75	27.86	6.53	0.11	0.001

Legend: TD, typical development; SpLD, specific learning disorder; ASD, autism spectrum disorder; CI, chronic illness.

**Table 4 jcm-12-00475-t004:** Scores of parental perception of children’s EF.

Children EFs	TD (*n* = 587)		SpLD (*n*= 23)		ASD (*n* = 9)		CI (*n* = 29)		Statistics	
	Mean	SD	Mean	SD	Mean	SD	Mean	SD	*F* Value	η^2^_p_
Working memory	12.72	3.30	11.18	2.40	10.56	3.54	12.44	3.19	2.81 *	0.013
Attention	12.27	3.01	11.00	2.35	9.56	3.36	12.10	2.44	3.96 **	0.018
Planning	11.64	3.02	10.87	3.05	9.78	3.11	11.76	2.38	1.91	0.009
Shifting	12.43	3.22	11.39	2.50	10.00	3.87	12.14	3.01	2.72 *	0.013
Inhibitory control	12.27	3.21	11.65	2.89	10.22	3.90	12.45	2.32	1.56	0.007

** *p* < 0.01, * *p* < 0.05. Legend: TD, typical development; SpLD, specific learning disorder; ASD, autism spectrum disorder; CI, chronic illness.

**Table 5 jcm-12-00475-t005:** Correlations among parental satisfaction, parental self-efficacy, parents’ perception of children’s EF, and parenting resources.

	1	2	3	4
1. Parental satisfaction	-	0.67 **	0.44 **	0.20 **
2. Parental self-efficacy	0.67 **	-	0.37 **	0.09 *
3. Perception of children’s EF	0.44 **	0.37 **	-	0.32 **
4. Parenting resources	0.20 **	0.09 *	0.32 **	-

** *p* < 0.01, * *p* < 0.05. Legend: TD, typical development; SpLD, specific learning disorder; ASD, autism spectrum disorder; CI, chronic illness.

## Data Availability

Data are available under request.

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
