# Peer review of "Parental Resources in Parents of Children with Special Needs (SNs) at the Time of COVID-19"

_jcm, 2023, doi:10.3390/jcm12020475_

Round 1

Reviewer 1 Report

The authors describe a cross-sectional survey on parental resources, satisfaction, self-efficacy and parents’ perception of their children’s executive functions by Italian parents during the Cov-19 pandemic. They examine differences in the mentioned variables between parents of typical developing children and three groups of children with special needs. Using a path analysis model the authors examine the impact of satisfaction, self-efficacy and type of special needs on parental resources and their perception of children’s executive functions. The study addresses the highly relevant topic of pandemic parental resources in managing children with the most common childhood developmental and behaviour disorders.

Results demonstrates that parents of children with specific learning disorders report the lowest parental resources.

However, there are some theoretical and methodological concerns and uncertainties that should be addressed before I can recommend the study for publication. My major concern is related to the reported executive functions. spLD, ADHD and ASD are disorders related to impairments also in executive functioning, even if to varying degrees. Thus, independent of Cov-19 pandemic, it is to be expected that parents would score the executive functions of these children lower. It is strongly recommended that the authors address this in the introduction and discussion and take this into account when interpreting the results. As no pre-pandemic data on executive functions (but also parental resources and sense of competence) seems to be available, it is difficult to draw conclusions about the effect of the pandemic situation itself. The authors should be careful with such conclusions (e.g. in the abstract). If there is a way to access pre-pandemic data, authors should certainly add this.

Here are my further concerns:

Methods

I miss information about the situation of the families during the data collection. Was it at the beginning of the pandemic, were schools still closed during the data collection, what other restrictions did the families face during the data collection (e.g. closed playgrounds, curfews)?

It is not clear how the children's diagnoses were captured. Presumably by parental report. This should be described by the authors (e.g. open question or predefined selection).

It is not clear why children with ADHD are grouped together with children with specific learning disorders and then referred to as the spLD group. These two disorders differ significantly in their symptoms. Especially with regard to executive functions, there are differences between the disorders. The authors need to explain the reason for combining these two disorders into one group and they should then label this group unambiguously. In addition, how were children with comorbid disorders, e.g., ADHD and ASD, assigned (or were there no such children)?

Results

The authors should add information on socio-demographic data for the four investigated groups separately and check whether the groups differ in terms of socio-demographics.

Please add effect sizes for the MANOVAS

Line 272 and figure 1: While in the text the authors describe the relationship with “type of pathology” in figure 1 it is named “presence of pathology”. The terms should be used clear and consistent.

Discussion

Line 319: self-efficacy in children or parents?

Line 342/343 “In this way, the study fills a gap in literature concerning children disability and in particular, intellectual disability and special needs” The investigated disorders (ADHD, ASD spLD) are no intellectual disability. The authors should describe this precisely.

At the methods section the authors described that “participants worked primarily as teacher” what makes the sample special. This should be taken up in the discussion - how it may have influenced the results if the parents themselves work as teachers.

Limitation

The authors should add to the limitations that it is difficult to draw conclusions about the effect of the pandemic situation itself as no pre-pandemic data was analyzed.

Minors:

The Abstracts includes many abbreviations partly explained much later in the text (e.g. CI). It is therefore hard to follow the aim and results of the study during the abstract.

Author Response

Dear Reviewer,

we would like to thank you for your insightful comments. We have provided a point by point answer to your comments. 

  • My major concern is related to the reported executive functions. spLD, ADHD and ASD are disorders related to impairments also in executive functioning, even if to varying degrees. Thus, independent of Cov-19 pandemic, it is to be expected that parents would score the executive functions of these children lower. It is strongly recommended that the authors address this in the introduction and discussion and take this into account when interpreting the results. As no pre-pandemic data on executive functions (but also parental resources and sense of competence) seems to be available, it is difficult to draw conclusions about the effect of the pandemic situation itself. The authors should be careful with such conclusions (e.g. in the abstract). If there is a way to access pre-pandemic data, authors should certainly add this.

Dear Reviewer, we understand and agree to your comments. We added this point to the introduction, to the discussion, and as limitation. Our intent, by the way, is also to describe the configuration of the relationships during the pandemic, that we imagine it could have worsened the dynamics, but our intent is not to compare before and after. We just assessed these dynamics as we found them when we spread the survey.

  • I miss information about the situation of the families during the data collection. Was it at the beginning of the pandemic, were schools still closed during the data collection, what other restrictions did the families face during the data collection (e.g. closed playgrounds, curfews)?

The information was collected at the end of the first wave of pandemic, that, in Italy, was very restrictive. The most restrictive measure that families faced was total lockdown: people were not allowed to go out for any reason, apart from groceries, where it was allowed to buy food and first necessity products only, and to do some distanced physical activity. The police controlled that all the people going out had a valid reason for it, through a written self-declaration that they would have registered in case of check.

  • It is not clear how the children's diagnoses were captured. Presumably by parental report. This should be described by the authors (e.g. open question or predefined selection).

Dear Reviewer, we add this information in the text. You are right, the diagnosis was reported by parents in the question “does your child have one or more of the following conditions?”.

It is not clear why children with ADHD are grouped together with children with specific learning disorders and then referred to as the spLD group. These two disorders differ significantly in their symptoms. Especially with regard to executive functions, there are differences between the disorders. The authors need to explain the reason for combining these two disorders into one group and they should then label this group unambiguously. In addition, how were children with comorbid disorders, e.g., ADHD and ASD, assigned (or were there no such children)?

 Dear Reviewer, actually we referred to one single case that displayed both ADHD and SpLD together. No other cases of multimorbidity were observed, therefore we decide to remove ADHD.

  • The authors should add information on socio-demographic data for the four investigated groups separately and check whether the groups differ in terms of socio-demographics.

Dear Reviewer, we added a table with the requested information.

  • Please add effect sizes for the MANOVAS

Dear Reviewer, we added the requested information.

  • Line 272 and figure 1: While in the text the authors describe the relationship with “type of pathology” in figure 1 it is named “presence of pathology”. The terms should be used clear and consistent.

Dear Reviewer, thanks for the comment. We chose to use “presence of pathology”.

  • Line 319: self-efficacy in children or parents?

Dear Reviewer, we referred to self-efficacy in parents and severity of disability/disorder in children. We slightly modified the sentence to be clearer.

  • Line 342/343 “In this way, the study fills a gap in literature concerning children disability and in particular, intellectual disability and special needs” The investigated disorders (ADHD, ASD spLD) are no intellectual disability. The authors should describe this precisely.

Dear Reviewer, you are right. We removed the reference to intellectual disability.

  • At the methods section the authors described that “participants worked primarily as teacher” what makes the sample special. This should be taken up in the discussion - how it may have influenced the results if the parents themselves work as teachers.

Dear Reviewer, we conducted the MANOVA again controlling for the occupation and no significant interactions between parents' job and childrens' condition emerged. Therefore, we added this information in the results section and in the discussion. 

  • The authors should add to the limitations that it is difficult to draw conclusions about the effect of the pandemic situation itself as no pre-pandemic data was analyzed.

Dear Reviewer, we discussed this point in the limitation section, as you suggested.

  • The Abstracts includes many abbreviations partly explained much later in the text (e.g. CI). It is therefore hard to follow the aim and results of the study during the abstract.

Dear Reviewer, we put the extended names avoiding acronyms, as suggested.

Thank you for your precious suggestions.

Reviewer 2 Report

I applaud the authors for conducting an interesting article with much potential, but it will need minor revisions in order to be published.
- Abstract: Efs; TD; SpLD; ASD; CI?; Line 106 – SES?; Line 201 – CI? - Explain before introducing the acronym.

- Why did the authors not consider the sociodemographic characteristics of the parents in the analyzes carried out? We know that, for example, the level of education may influence the perceptions of the main variables. Does the model maintain its stability when controlling for sociodemographic variables?

Author Response

Dear Reviewer,

Thank you for the insightful comments. We answered to your concerns.

- Abstract: Efs; TD; SpLD; ASD; CI?; Line 106 – SES?; Line 201 – CI? - Explain before introducing the acronym.

Thanks for the comment. We reported the terms in the extended form.

- Why did the authors not consider the sociodemographic characteristics of the parents in the analyzes carried out? We know that, for example, the level of education may influence the perceptions of the main variables. Does the model maintain its stability when controlling for sociodemographic variables?

Thank you for this comment. Actually, we already tried to run the analyses controlling for level of education, but it has no influence at all. By the way, we put this in detail within the analyses and reported more descriptive statistics about sociodemographic data.

Thank you again for your precious suggestions!

Round 2

Reviewer 1 Report

I am pleased to review the revision of the manuscript.

The authors have responded to and edited all comments and have addressed concerns appropriately. I still have 2 minor concerns.  

1. It would be more informative if in the crosstab the percentages per column (i.e. per development condition) are given and not of the entire table (i.e. the total sample). This would make it easier to see the differences between the groups. For example, what percentage of parents of TD children have a middle school diploma compared to parents of children with ASD. 

2. Line 356-360: I think "decrease" is not the appropriate term since two groups are being compared and not a change over time. It should be better described that parents of children with spLD observe lower working memory and attention than parents of TDchildren. The same applies to the sentence about parents of children with ASD 

Author Response

Dear Reviewer, 

thank you so much for your comments. 

1. We replaced the column percentage, as you required. 

2. We rephrased the sentence according to your suggestion.

Thank you so much for your useful comments.